# National Development in the Use of Inhaled Corticosteroid Treatment in Chronic Obstructive Pulmonary Disease: Repeated Cross-Sectional Studies from 1998 to 2018

**DOI:** 10.3390/biomedicines12020372

**Published:** 2024-02-05

**Authors:** Allan Klitgaard, Rikke Ibsen, Jesper Lykkegaard, Ole Hilberg, Anders Løkke

**Affiliations:** 1Department of Regional Health Research, University of Southern Denmark, 5230 Odense, Denmark; ole.hilberg@rsyd.dk (O.H.); anders.lokke@rsyd.dk (A.L.); 2Department of Internal Medicine Vejle, University Hospital of Southern Denmark, 7100 Vejle, Denmark; 3i2minds, 8000 Aarhus, Denmark; rikke@i2minds.dk; 4Research Unit of General Practice, Department of Public Health, University of Southern Denmark, 6705 Esbjerg, Denmark; jlykkegaard@health.sdu.dk

**Keywords:** chronic obstructive pulmonary disease, inhaled corticosteroids, inhaled medication, pharmacoepidemiology, treatment recommendations, nationwide development

## Abstract

Recommendations for the treatment of chronic obstructive pulmonary disease (COPD) have shifted towards a more restrictive use of inhaled corticosteroids (ICS). We aimed to identify the nationwide development over time in the use of ICS treatment in COPD. We conducted a register-based repeated cross-sectional study using Danish nationwide registers. On a yearly basis from 1998 to 2018, we included all patients in Denmark ≥ 40 years of age with an ICD-10 diagnosis of COPD (J44). Accumulated ICS use was calculated for each year based on redeemed prescriptions. Patients were divided into the following groups: No ICS, low-dose ICS, medium-dose ICS, or high-dose ICS. From 1998 to 2018, the yearly proportion of patients without ICS treatment increased (from 50.6% to 57.6%), the proportion of patients on low-dose ICS treatment increased (from 11.3% to 14.9%), and the proportion of patients on high-dose ICS treatment decreased (from 17.0% to 9.4%). We demonstrated a national reduction in the use of ICS treatment in COPD from 1998 to 2018, with an increase in the proportion of patients without ICS and on low-dose ICS treatment and a decrease in the proportion of patients on high-dose ICS treatment.

## 1. Introduction

Chronic obstructive pulmonary disease (COPD) is a chronic condition characterized by the irreversible destruction of lung tissue, which causes chronic respiratory symptoms and persistent airflow obstruction [1]. It is commonly regarded as one of the biggest health problems worldwide [2,3]. The disease itself is incurable, and treatment is centered on relieving symptoms and preventing complications and disease progression [1].

The pharmacological treatment of COPD mainly consists of inhaled medication, and inhaled corticosteroid (ICS) treatment is one such widely used treatment [1,4]. In general, inhaled medications are symptom-relieving and preventive against disease progression and complications, such as acute exacerbations [5,6,7]. ICS-containing medication has especially proven effective in preventing exacerbations [1,8], and the effectiveness of ICS in certain COPD phenotypes has been established. In patients with eosinophil COPD [5,9,10,11,12,13,14], asthmatic features [1,15,16], or frequent exacerbations [17], ICS treatment has been shown to decrease the risk of exacerbations. ICS has been shown to increase the quality of life among COPD patients [18,19,20], and there is evidence suggesting a reduction in mortality in selected patients [21,22].

Despite the positive effects, extensive use of ICS in COPD has been cautioned against during the past decades [4,23,24,25,26,27,28,29], and the Global Initiative for Chronic Lung Disease (GOLD), which is a world-leading authority within COPD management, has published increasingly restrictive ICS treatment recommendations [1]. This is largely due to the negative side effects associated with the treatment [30] and the fact that only subpopulations of COPD patients have the benefits [1]. Despite these warnings, several studies have demonstrated a high usage of ICS, with an estimated 50–80% of all COPD patients in ICS treatment [31,32,33,34]. A recent Danish study found that 39–55% of patients with GOLD stage B were treated with ICS-containing medication [35].

The development of ICS treatment in COPD in Denmark has not been studied before. A recent nationwide study has identified an increase in the overall number of redeemed prescriptions of ICS-containing medication in Denmark during the past two decades [36], but the authors did not have information about the indication for prescribing the inhaled medication. Our current study therefore aimed to investigate the nationwide use of ICS in patients with a hospital-registered COPD diagnosis in Denmark during the past two decades.

## 2. Materials and Methods

### 2.1. Data Sources

This study was based on Danish nationwide registers. Data were collected from the Civil Registration System (CRS), the Danish National Patient Register (DNPR), the Danish National Prescription Register (NPR), and Statistics Denmark (DST). Individual-level linkage across all registers was accomplished using the unique personal identification number that is assigned to every citizen in Denmark.

### 2.2. Study Design and Population

We conducted annual cross-sectional studies from 1998 to 2018. For each year, we included all patients in Denmark with an International Classification of Diseases 10th Revision (ICD-10) diagnosis code of COPD (J44) registered in the DNPR who were alive on the 31st of December in the year of study. To be included, the patients had to be ≥40 years of age at the time of diagnosis. All patients in Denmark are assigned one or more ICD-10-coded diagnoses upon each in- or outpatient hospital encounter. We had access to data from 1995 to 2018.

### 2.3. Inhaled Corticosteroid Treatment

The main variable of interest in this study was ICS treatment, which was collected from the NPR. ICS-containing medication is sold in Denmark only with a prescription. A patient was considered for ICS treatment if they had redeemed two or more prescriptions of any ICS-containing medication within the year of interest. Anatomical Therapeutic Chemical (ATC) codes are shown in Table 1. The daily ICS dose was calculated as the average daily exposure during each year of the study. Like previous studies on ICS treatment [37,38,39], ICS doses were calculated using standard-particle beclomethasone dipropionate equivalents estimated based on the National Institute for Health and Care Excellence ICS dose chart [40]. Based on accumulated ICS dose during each year of study, patients were grouped by average daily ICS dose as follows: No ICS, low-dose ICS (<500 micrograms daily), medium-dose ICS (500–1200 micrograms daily), or high-dose ICS (>1200 micrograms daily) [37,40]. A detailed description of ICS treatment calculations can be found in the Appendix A.

### 2.4. Other Variables

Age and sex were collected from the CSR. Data on socio-economy were collected from various registers at DST (co-habitation status, income status, and highest completed education). Data on comorbidity were collected from the DNPR using ICD-10 diagnosis codes, from which we constructed the Charlson Comorbidity Index (CCI) as described by Quan et al. [41]. We saw a large decrease in registered comorbidities during the last three years due to a delay in the reporting of ICD-10 diagnosis codes in the DNPR, and we decided to include comorbidities only until 2015. Data on comorbidities for all years can be seen in Appendix A.

### 2.5. Statistical Analysis

We calculated the proportion of patients in each ICS treatment group for each year of study. We also calculated the proportion of patients in each group of age (40–49 years, 50–59 years, 60–69 years, 70–79 years, or 80+ years), sex (male or female), CCI (0, 1, 2, or 3+), co-habitation status (living alone or co-habiting), education (primary, secondary, vocational, or college), and income status (employed, unemployed, disability pension, early retirement, age pension, education, or other). For age, we also calculated the mean and standard deviation. All analyses were performed using SAS 9.4 TS Level 1M5 (SAS, Inc., Cary, NC, USA).

## 3. Results

### 3.1. Study Population

Study population characteristics from 1998, 2008, and 2018 can be seen in Table 2, displayed as absolute numbers and frequencies. The development of these characteristics for each year from 1998 to 2018 is shown in Figure 1a,b. Data for each year in the study period can be seen in Appendix A.

The study population with a COPD diagnosis was 35.565 patients in 1998 and 99.057 patients in 2018. The population in 2018 was older than the population in 1998, with a mean age of 72 years vs. 69 years. An increase in the proportion of patients over 80 years of age was especially seen (24.5% in 2018 vs. 16.1% in 1998). Overall, the distribution of patients according to the CCI groups was similar between 1998 and 2015. Changes from 1998 to 2018 were seen regarding socio-economic variables: compared to 1998, a larger proportion of patients in 2018 lived alone (52.5% vs. 48.0%), while a larger proportion received income from age pension (71.8% vs. 60.3%) and a smaller proportion received income from disability pension (10.5% vs. 20.2%) and early retirement (1.8% vs. 6.5%).

### 3.2. Inhaled Corticosteroid Treatment

The proportion of patients according to ICS treatment groups in 1998, 2008, and 2018 is shown in Table 3. From 1998 to 2018, the proportion of patients not in ICS treatment increased from 50.6% to 57.6%. The proportion of patients in low-dose ICS treatment increased (11.3% to 14.9%), while there was a decrease in the proportion of patients in medium-dose ICS treatment (21.1% to 18.1%) and high-dose ICS treatment (17.0% to 9.4%). The development of ICS treatment, including doses, for each year from 1998 to 2018 is shown in Figure 2. The decline in medium-dose and high-dose ICS treatment showed a stable trend during the entire study period. The increase in the proportion of patients not in ICS treatment was less stable, with a decrease in this proportion from 2002 to 2009 and a corresponding increase in the proportion of patients on medium-dose ICS treatment. ICS treatment data for each year in the study period can be seen in Appendix A.

## 4. Discussion

We demonstrated a nationwide reduction from 1998 to 2018 for the use of ICS treatment per patient with an ICD-10 diagnosis code of COPD.

This general reduction in ICS use may partly be explained by shifts in international recommendations. The first GOLD report was published in 2001 [42,43], and it was revised in 2006 [44], 2011 [45], and 2017 [46]. A new report has been published each year since 2017. In the 2001 report, the classification of COPD severity was based solely on lung function, and ICS treatment was recommended in patients with forced expiratory volume in the first second (FEV1) < 80% of predicted lung function response to glucocorticosteroids [43]. The classification of COPD severity remained based on lung function in the 2006 report [44], but the use of ICS was now only recommended in patients with FEV1 < 50% of predicted and repeated exacerbations. In the 2011 report, COPD severity and treatment recommendations were now, for the first time, based on both lung function and the individual patient’s symptoms and future risk of exacerbations [45]. The recommendations for ICS treatment were the same as in the 2006 report, except only in patients not already adequately controlled by long-acting bronchodilator therapy. The 2017 report recommendations for pharmacological treatment choices were only based on exacerbation history and symptom burden, while lung function was not recommended to guide clinicians in this aspect [46]. Treatment with dual bronchodilator therapy was now recommended over ICS-containing therapy as the initial treatment choice in patients with high exacerbation risk. In short, leading international evidence-based recommendations during the past two decades have become increasingly strict regarding the use of ICS in COPD. One thing, however, is the publication of guidelines, but whether these guidelines are followed is another issue. Studies in the last decade have revealed a high use of ICS by both primary care practitioners (PCP) [47,48,49,50,51] and pulmonary specialists [34,51,52,53]. Although some studies have indicated a prescription pattern in accordance with treatment guidelines [54,55], most studies have mainly focused on poor adherence to guidelines [48,51,52,56,57,58]. In 2015, Davis et al. reported a study that investigated physicians’ knowledge and implementation of GOLD treatment recommendations, where they found the proportion of PCPs and pulmonary specialists providing treatment in concordance with recommendations to be between 38% and 67% depending on GOLD disease severity stage [59]. They also found that 58% of PCPs reported awareness of GOLD recommendations compared to 93% of pulmonary specialists. A study by Sulku et al. from 2019 found that 55% of patients with COPD on ICS treatment met the criteria for recommended discontinuation [51]. A recent study by Alabi et al. found that 19% of patients in GOLD group A were treated in concordance with GOLD recommendations compared to 26% in GOLD group B and 92% in GOLD group D. In summary, a vast body of evidence indicates a less-than-ideal adherence to treatment recommendations. However, adherence to recommendations is a two-sided coin: when looking at the 38–67% not adhering to recommendations [59], it is easy to forget the 33–42% who are and that this 33–42% does have an impact on the larger scale. Therefore, we believe it likely that the increasingly strict treatment recommendations have influenced the use of ICS in COPD in Denmark. This is supported by a Danish study from 2010 where GPs’ adherence to guidelines, including prescription of ICS in patients with mild COPD, was improved significantly after information on guideline-recommended treatment [60].

Interestingly, we saw a decrease from 2002 to 2009 in the proportion of patients not in ICS treatment, whereas the proportion of patients in low-dose ICS increased. In this period, the proportion of patients in medium and high-dose ICS was still declining. This suggests that a shift from “no ICS” to “low-dose ICS” happened in this period, and the reasons for this can only be speculated upon. During the early 2000s, emerging evidence of survival benefits of adding ICS to LABA treatment was published [61,62,63], and the first randomized controlled trial to demonstrate significant beneficial effects of ICS/LABA combination therapy in patients with COPD was published in 2007 [64,65]. This, in combination with the authorization of the first ICS/LABA single inhaler combination in 1999 [36], may be responsible for this shift from “no ICS” to “low-dose ICS”. After 2009, however, we saw a marked increase in the proportion of patients not in ICS treatment. In this period, there was an increasing research focus on the beneficial effects of bronchodilator therapy without ICS [19,66,67,68], leading up to two landmark studies: the FLAME study in 2016 that showed dual bronchodilator therapy to be equally effective in preventing exacerbations than ICS-bronchodilator combination therapy [6], and the WISDOM study in 2014 indicating that selected patients may discontinue ICS therapy without increased risk of adverse events [69]. Additionally, the first dual bronchodilator single inhaler combination was authorized in 2013 [70,71]. This, combined with guidelines, may explain the increasing proportion of patients not in treatment with ICS since 2009. Furthermore, Reilev et al. demonstrated a nationwide increase in bronchodilator use in Denmark from 2000 to 2016 [36].

We documented differences in study population characteristics from 1998 to 2018: the population in 2018 was slightly older than the population in 1998, and the proportion of patients over 80 years of age has especially increased. Because disease severity is likely to increase with advancing age [72], and ICS use increases with disease severity, this age increase in our study population would likely increase the use of ICS as time advanced.

Our study has some limitations. Our study was conducted as a repeated cross-sectional study with retrospective data retrieval for each year. This excludes patients who die during the year of study, and these are likely the patients with the most severe disease and more likely to be in medium-to-high-dose ICS treatment. This approach therefore has the potential to underestimate the proportion of patients in medium-to-high-dose ICS treatment, which is important to consider when comparing our findings with other studies. However, the goal of our study was to only investigate the development over a large time-period, and we favored this simple and easily interpretable approach. Conversely, our study population selection may have the potential to overestimate the proportion of patients in ICS treatment because we included patients with an ICD-10 diagnosis code of COPD. Patients with COPD handled entirely by their PCP are invisible to us via registers because they do not have a registered ICD-10 diagnosis code. These are likely the patients with mild disease and therefore not in ICS treatment [47,52,73]: a recent study has shown that about 32% of patients with COPD in Denmark handled only by their PCP were in ICS treatment, which is considerably lower than in our study [74]. The severity threshold for COPD hospital admission and ICD-10 code registration has increased during our study period [75], which would further increase this potential bias over time. However, the study by Lykkegaard et al. did not include outpatient hospital encounters. The reduction in ICS use in our study was mainly driven by a large increase in the number of patients not in ICS treatment, which may be explained by an increased overall focus on the disease, leading to a larger proportion of patients with mild disease being assessed in a hospital outpatient setting. Supporting this theory is the marked growth from 1998 to 2018 in the population of patients with a hospital-registered COPD diagnosis, which cannot be explained by general population growth in Denmark, as this was under 1% yearly for the study period [76]. The confinement to pre-defined data is a limitation of register-based research, and we are unable to determine the potential causal relationship between clinical characteristics and ICS use. However, we consider this to be outside the scope of our study, which was designed to be purely descriptive. The cross-sectional design does not allow for the investigation of changes in ICS treatment at the individual level, which was intentional as the scope of our study was about changes at the population level. Strengths of our study include the long study period and the large nationwide study population derived from national registers, which ensures limited selection bias. The validity of these registers was investigated and judged high [77,78].

## 5. Conclusions

We demonstrated a nationwide decrease in the proportion of patients with a hospital-registered COPD diagnosis in ICS treatment from 1998 to 2018, and a decrease in the ICS dose used. This may reflect a combination of the emergence of evidence towards the beneficial effects of non-ICS-containing medication, an increase in non-ICS-containing pharmacological treatment options in COPD, and increasingly restricted recommendations for ICS therapy in international evidence-based guidelines.

## Figures and Tables

**Figure 1 biomedicines-12-00372-f001:**
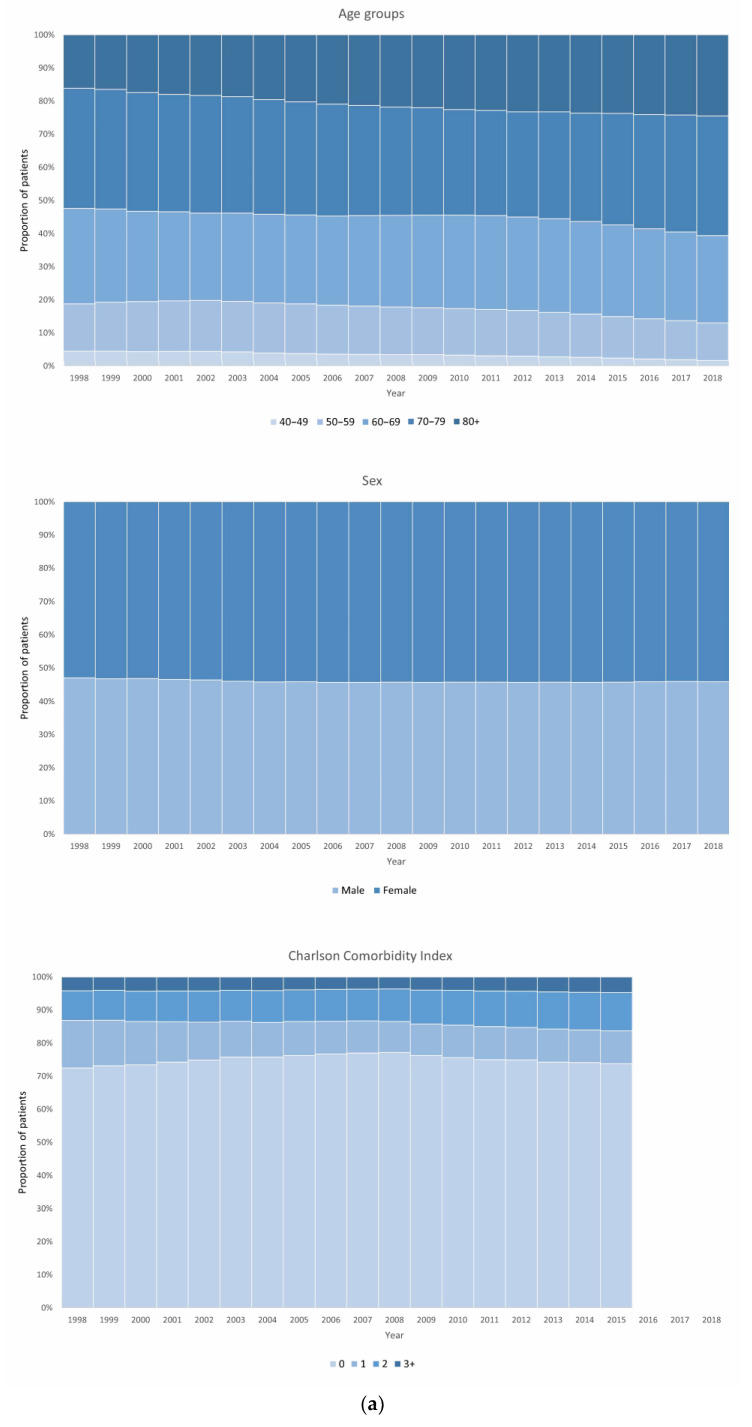
(**a**) Annual study population characteristics (age groups, sex, and comorbidities). (**b**) Annual study population characteristics (co-habitation status, education, and income status). * “Education” and “other” constitute < 1% each year.

**Figure 2 biomedicines-12-00372-f002:**
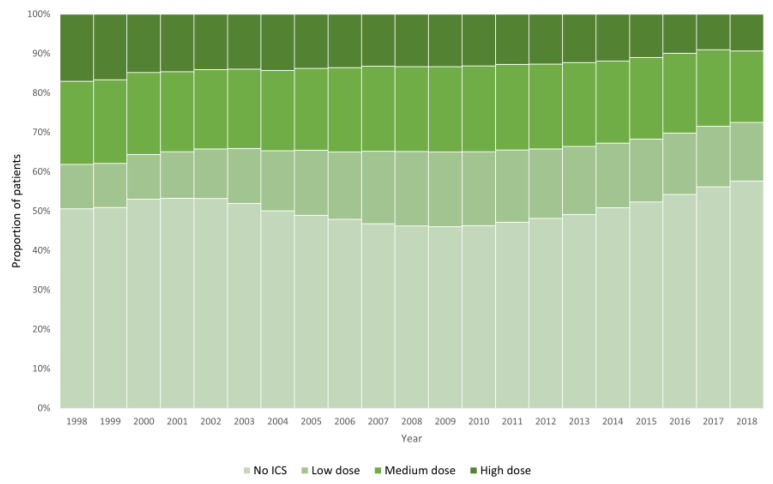
Annual proportion of patients in treatment with ICS (including dose) from 1998 to 2018.

**Table 1 biomedicines-12-00372-t001:** ATC codes used for identifying inhaled corticosteroid treatment.

Medication	ATC Codes
ICS	R03BA [01–09]
ICS + LABA	R03AK [06–14]
ICS + LABA + LAMA	R03AL08, R03AL09, R03AL11, R03AL12

Abbreviations: ATC, anatomical therapeutic chemical. ICS, inhaled corticosteroid. LABA, long-acting beta-2-agonist. LAMA, long-acting muscarinic agonist.

**Table 2 biomedicines-12-00372-t002:** Characteristics of study population in the years 1998, 2008, and 2018.

	1998		2008		2018	
Population (*n*)	35,565		77,314		99,057	
	*n*	%	*n*	%	*n*	%
**Sex**						
Male	16,716	47.0	35,348	45.7	45,396	45.8
Female	18,849	53.0	41,966	54.3	53,661	54.2
**Age, mean (SD)**	69	(10.5)	70	(10.9)	72	(10.4)
**Age group**						
40–49	1602	4.5	2641	3.4	1674	1.7
50–59	5090	14.3	11,125	14.4	11,239	11.3
60–69	10,243	28.8	21,409	27.7	26,170	26.4
70–79	12,895	36.3	25,287	32.7	35,745	36.1
80+	5735	16.1	16,852	21.8	24,229	24.5
**Charlson Comorbidity Index ***						
0	25,790	72.5	59,668	77.2	72,139 *	73.8 *
1	5130	14.4	7245	9.4	9770 *	10.0 *
2	3170	8.9	7590	9.8	11,244 *	11.5 *
3+	1475	4.1	2811	3.6	4628 *	4.7 *
**Co-habitation status**						
Living alone	17,069	48.0	39,169	50.7	52,032	52.5
Co-habiting	18,496	52.0	38,145	49.3	47,025	47.5
**Education**						
Primary	17,698	49.8	40,857	52.8	45,484	45.9
Secondary	196	0.6	875	1.1	1687	1.7
Vocational	7298	20.5	23,056	29.8	35,681	36.0
College **	2009	5.6	7567	9.8	13,429	13.6
Missing	8364	23.5	4959	6.4	2776	2.8
**Income type**						
Employed	3435	9.7	9867	12.8	10,868	11.0
Unemployed (primarily)	883	2.5	1712	2.2	4294	4.3
Disability pension	7194	20.2	10,865	14.1	10,357	10.5
Early retirement	2302	6.5	3287	4.3	1754	1.8
Age pension	21,429	60.3	51,091	66.1	71,151	71.8
Education	-	-	10	<0.1	29	<0.1
Other ***	319	0.9	482	0.6	604	0.6

* Data on comorbidities are from 2015; see explanation in text. ** Short college, medium college, and Master’s/Ph.D. summed. *** denotes 1 observation missing information on income type; this is included in “Other”. Abbreviations: SD, standard deviation.

**Table 3 biomedicines-12-00372-t003:** Proportion of patients in treatment with ICS (including dose) in the years 1998, 2008, and 2018.

	1998		2008		2018	
Population (*n*)	35,565		77,314		99,057	
	* **n** *	**%**	* **n** *	**%**	* **n** *	**%**
**ICS treatment ***						
No ICS	17,986	50.6	35,734	46.2	57,072	57.6
Low dose	4009	11.3	14,612	18.9	14,782	14.9
Medium dose	7508	21.1	16,639	21.5	17,931	18.1
High dose	6053	17.0	10,319	13.3	9272	9.4

* Observations with missing information are not included in the ICS distribution. Therefore, sum is not always = 100%.

## Data Availability

Data from registers are not publicly available. The data supporting the conclusions of this article are available from registers upon request and approval of access by national authorities.

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
