# Peer review of "National Development in the Use of Inhaled Corticosteroid Treatment in Chronic Obstructive Pulmonary Disease: Repeated Cross-Sectional Studies from 1998 to 2018"

_biomedicines, 2024, doi:10.3390/biomedicines12020372_

Round 1

Reviewer 1 Report

Comments and Suggestions for Authors

In the manuscript an important topic is discussed, it is interesting to see how the guidelines met the “reality”. Some suggestions to improve the quality of the manuscript:

·         Row 73-74: ”We had access to data from 1995 to 1998.” We do not have access to data .... ?

·         Row 78: “prescriptions of ICS-containing medication” – it is not clearly defined if ICS containing fixed-dose combination are included or not.

·         In the Material and Method section should be mentioned if it is allowed in Denmark to by ICS without prescription (leaving no footprint in the system)

·         Verifying the classification of low, medium and high doses of ICS (according to the manuscript data: low dose ICS (<500 micrograms daily), medium dose ICS (500-1200 micrograms daily), high dose ICS (>1200 micrograms daily): in the reference nr 37 these data are not available. Please introduce a relevant reference.

·         Row 86: „daily).[37]” correctly „daily) [37].”

·         Results section: Data in a figure that contain the number of the patients for each year is recommended

·         Can be mentioned, that Denmark population growth rate – in the tested period - was below 1%, so it is not an influencing factor.

·         At the limitation of the study can be mentioned the fact that it is a cross-sectional study not a longitudinal one, therefore changes at the individual level was not possible to investigate.

Author Response

Dear Reviewer

Thank you very much for your review – we believe that it has improved the manuscript. Below is a point-by-point answer with reviewer comments in blue text.

In the manuscript an important topic is discussed, it is interesting to see how the guidelines met the “reality”. Some suggestions to improve the quality of the manuscript:

Row 73-74: ”We had access to data from 1995 to 1998.” We do not have access to data .... ?

This is a mistake, thank you. It has been corrected. We had access to data from 1995 to 2018.

Row 78: “prescriptions of ICS-containing medication” – it is not clearly defined if ICS containing fixed-dose combination are included or not.

Thank you for pointing towards this. We want the methods to be as clearly described as possible. We have changed the sentence to: “...prescription of any ICS-containing medication...”

For full transparency, all ATC codes are displayed in Table 1, and the relevant drugs may be found here: https://www.whocc.no/atc_ddd_index/?code=R03&showdescription=yes

In the Material and Method section should be mentioned if it is allowed in Denmark to by ICS without prescription (leaving no footprint in the system)

This is a simple, yet very important, observation. Thank you. It has been added (Line 77).

Verifying the classification of low, medium and high doses of ICS (according to the manuscript data: low dose ICS (<500 micrograms daily), medium dose ICS (500-1200 micrograms daily), high dose ICS (>1200 micrograms daily): in the reference nr 37 these data are not available. Please introduce a relevant reference.

The definition of ICS is paramount to our study, so thank you for this insight. We believe reference 37 to be of vital importance in this regard, even though the paper does not explicitly state the dosage classification of ICS. Håkansson et al (reference 39) has done multiple studies using this approach, and one of those studies was Reference 37, which was made in collaboration with our study group and our statistician (Rikke Ibsen). For this reason, this is a study where we are one hundred percent certain that the ICS classification used was the same as our current study.

To make up for the missing data on classification in reference 37, we have changed the reference to the NICE ICS dose chart (reference 40) to a webpage-reference, so that a link to the dose chart is provided in the reference, and we have included this as a reference for the ICS dose classification as well.

Row 86: „daily).[37]” correctly „daily) [37].”

This has been corrected.

Results section: Data in a figure that contain the number of the patients for each year is recommended.

This is a good suggestion, and we would usually agree with this. We have considered including yearly study populations at the top in Figure 2 but decided not to because 1) the study populations for each year are large numbers, and the text would have to be unreadably small to fit within the bar of each year in the bar chart. 2) The added value of the information provided is not big enough to make up for this, in our opinion. 3) The growth of the COPD population is already well-documented in Table 2 and 3, and it is documented for each year in Table S3, Appendix A, for the readers who want more detailed knowledge.

Can be mentioned, that Denmark population growth rate – in the tested period - was below 1%, so it is not an influencing factor.

Interesting perspective, thank you. This has been added (lines 243-246).

At the limitation of the study can be mentioned the fact that it is a cross-sectional study not a longitudinal one, therefore changes at the individual level was not possible to investigate.

This has been added (lines 249-251).

Reviewer 2 Report

Comments and Suggestions for Authors

Dear Authors,

thank you for the opportunity to read and review the manuscript.

The topic is interesting and the article is overall well written.

Specific comments

Materials and methods

Why did you include just patients with more of 40 years of age at the time of the diagnosis? In 2023 GOLD report a wide spectrum of lung impairment has been reported, even in young patients.

There are just few studies on young COPD patients but I think that we need to detect the overall COPD population to improve the global outcome [Wang Z, Li Y, Lin J, Huang J, Zhang Q, Wang F, Tan L, Liu S, Gao Y, Peng S, Fang H, Weng Y, Li S, Gao Y, Zhong N, Zheng J. Prevalence, risk factors, and mortality of COPD in young people in the USA: results from a population-based retrospective cohort. BMJ Open Respir Res. 2023 Jul;10(1):e001550. doi: 10.1136/bmjresp-2022-001550. PMID: 37451700; PMCID: PMC10351298.]

Results

Results (table 2, line 122-142, table 3) are interesting but not supported by statistical analysis that will make these data reliable and of scientific relevance. Statistical tests should be performed. Figure 1 is interesting. 

Are there data on the severity of COPD in the patients included in the study? This could be of relevance, as you stated in line 222-223.

Paragraph from line 221 to 248 has a different format.

Author Response

Dear Reviewer,

Thank you for reviewing our manuscript and helping us improve upon our work. Below is a point-by-point answer to the concerns raised, with reviewer’s comments in blue text.

Dear Authors,

Thank you for the opportunity to read and review the manuscript.

The topic is interesting, and the article is overall well written.

Specific comments

Materials and methods

Why did you include just patients with more of 40 years of age at the time of the diagnosis? In 2023 GOLD report a wide spectrum of lung impairment has been reported, even in young patients.

There are just few studies on young COPD patients but I think that we need to detect the overall COPD population to improve the global outcome [Wang Z, Li Y, Lin J, Huang J, Zhang Q, Wang F, Tan L, Liu S, Gao Y, Peng S, Fang H, Weng Y, Li S, Gao Y, Zhong N, Zheng J. Prevalence, risk factors, and mortality of COPD in young people in the USA: results from a population-based retrospective cohort. BMJ Open Respir Res. 2023 Jul;10(1):e001550. doi: 10.1136/bmjresp-2022-001550. PMID: 37451700; PMCID: PMC10351298.]

Thank you for pointing towards this important issue of study population selection. While we agree that detecting the overall COPD population is important, including patients under 40 years of age in this register-based study would cause more problems than it solved. Because COPD under the age of 40 is extremely rare (1), including patients under 40 years of age increases the likelihood of misclassification of asthma as COPD. As asthma patients are more likely to use inhaled corticosteroids, this would cause bias regarding our primary outcome.

Furthermore, this study population selection method is very common for defining COPD in register-based research in both Denmark and other countries (2-7).

Results

Results (table 2, line 122-142, table 3) are interesting but not supported by statistical analysis that will make these data reliable and of scientific relevance. Statistical tests should be performed. Figure 1 is interesting.

Thank you for raising the question of when and how to use statistical analysis.

There are some reasons why we have decided not to use statistical tests in our study. With study population sizes from approximately 35,000 to 100,000 patients, it is guaranteed that any small difference between groups will be statistically significant – but statistical significance does not necessarily mean clinically important. We have chosen to no use statistical testing to avoid taking focus from what is important: the clinical and overall significance of the development in nationwide ICS use.

Based on the above, we believe that the scientific relevance and data reliability of a research study does not always lie in statistical testing of descriptive data (8, 9).

if the Biomedicines Editor requires statistical testing of our results, we will of course comply.

Are there data on the severity of COPD in the patients included in the study? This could be of relevance, as you stated in line 222-223.

For this particular study, it is not possible.

For future studies, it is however possible to retrieve this information on a subset of patients from The Danish Register for COPD, but: 1) Patients are only included in this register upon contact with a hospital pulmonary outpatient clinic, and this would cause the study population to be considerably smaller. 2) The study population would likely not be representative of the total COPD population in Denmark. 3) The register has existed only from 2008 and onwards, so we would not have data for the entire study period.

Paragraph from line 221 to 248 has a different format.

This is true. We have used the official template of the journal, and we are unable to resolve this issue. Should the manuscript be accepted, we hope that the Biomedicines Editorial Office will be able to resolve this.

References

  1. Mannino DM. COPD: epidemiology, prevalence, morbidity and mortality, and disease heterogeneity. Chest. 2002;121(5 Suppl):121s-6s.
  2. Buhl R, Wilke T, Picker N, Schmidt O, Hechtner M, Kondla A, et al. Real-World Treatment of Patients Newly Diagnosed with Chronic Obstructive Pulmonary Disease: A Retrospective German Claims Data Analysis. Int J Chron Obstruct Pulmon Dis. 2022;17:2355-67.
  3. Waeijen-Smit K, Jacobsen PA, Houben-Wilke S, Simons SO, Franssen FME, Spruit MA, et al. All-cause admissions following a first ever exacerbation-related hospitalisation in COPD. ERJ Open Res. 2023;9(1).
  4. Liew CQ, Hsu SH, Ko CH, Chou EH, Herrala J, Lu TC, et al. Acute exacerbation of chronic obstructive pulmonary disease in United States emergency departments, 2010-2018. BMC Pulm Med. 2023;23(1):217.
  5. Bogart M, Germain G, Laliberté F, Lejeune D, Duh MS. Real-World Treatment Patterns and Switching Following Moderate/Severe Chronic Obstructive Pulmonary Disease Exacerbation in Patients with Commercial or Medicare Insurance in the United States. Int J Chron Obstruct Pulmon Dis. 2023;18:1575-86.
  6. Søgaard M, Madsen M, Løkke A, Hilberg O, Sørensen HT, Thomsen RW. Incidence and outcomes of patients hospitalized with COPD exacerbation with and without pneumonia. Int J Chron Obstruct Pulmon Dis. 2016;11:455-65.
  7. Lash TL, Johansen MB, Christensen S, Baron JA, Rothman KJ, Hansen JG, et al. Hospitalization rates and survival associated with COPD: a nationwide Danish cohort study. Lung. 2011;189(1):27-35.
  8. Wasserstein RL, Lazar NA. The ASA Statement on p-Values: Context, Process, and Purpose. The American Statistician. 2016;70(2):129-33.
  9. Hayes-Larson E, Kezios KL, Mooney SJ, Lovasi G. Who is in this study, anyway? Guidelines for a useful Table 1. J Clin Epidemiol. 2019;114:125-32.

Round 2

Reviewer 2 Report

Comments and Suggestions for Authors

Dear Authors, 

I read your answers with interest, thank you.